# Tidal Volume in Pediatric Ventilation: Do You Get What You See?

**DOI:** 10.3390/jcm11010098

**Published:** 2021-12-24

**Authors:** Erik Koomen, Joppe Nijman, Ben Nieuwenstein, Teus Kappen

**Affiliations:** 1Department of Pediatrics, Wilhelmina Children’s Hospital, University Medical Center Utrecht, 3508 AB Utrecht, The Netherlands; J.Nijman@umcutrecht.nl; 2Department of Medical Technology & Clinical Physics, University Medical Center Utrecht, 3508 GA Utrecht, The Netherlands; B.Nieuwenstein@umcutrecht.nl; 3Department of Anesthesia, Intensive Care and Emergency, University Medical Center Utrecht, 3508 GA Utrecht, The Netherlands; T.Kappen@umcutrecht.nl

**Keywords:** pediatric ventilation, technical challenges, software, clinical perception of technique

## Abstract

Mechanical ventilators are increasingly evolving into computer-driven devices. These technical advancements have impact on clinical decisions in pediatric intensive care units (PICUs). A good understanding of the design of mechanical ventilators can improve clinical care. Tidal volume (TV) is one of the corner stones of ventilation: multiple technical factors influence the TV and, thus, influence clinical decision making. Ventilator manufacturers make various design choices regarding the phase, site and conditions of TV measurement as well as algorithmic processing choices. Such choice may impact the measurement and subsequent display of TV. A software change of the TV measuring algorithm of the SERVO-i^®^ (Getinge, Solna, Sweden) at the PICU of the University Medical Centre Utrecht was studied in a prospective cohort. It showed, as example, a clinically significant impact of 8% difference in reported TV. Design choices in both the hardware and software of mechanical ventilators can have a clinically relevant impact on the measurement of tidal volume. In our search for the optimal TV for lung-protective ventilation, such choices should be taken into account.

## 1. Introduction

Mechanical ventilators are more and more frequently becoming computer-driven devices. The time of the iron lung or basic ventilators has passed and transitioned into an era of highly sophisticated ventilators on our pediatric intensive care units (PICUs). Multiple sensors and at least one computer processor is controlled by advanced software algorithms that deliver the breath air and oxygen that these devices deliver. The advances in ventilator technology raise the question whether we, as clinicians, still understand how we are ventilating our patients. Or, more specifically: do we really know what tidal volume we are administering to the lungs our patients, especially in our neonatal and infant populations?

In this manuscript we aim to provide more insight into this question. First, we give an expert overview of the different approaches that are being used in ventilator devices to measure and control tidal volumes that are being administered. Further, as research regarding the clinical impact of different solutions is lacking, we performed a small prospective cohort study to illustrate whether a software change in the ventilators at our PICU affected the administered tidal volumes in our patients.

## 2. Tidal Volume in Pediatric Ventilation: Technical Challenges and Clinical Consequences

Control of the tidal volume is of critical importance in mechanical ventilation. On the one hand the tidal volume needs to be sufficiently large to ventilate the alveoli, deliver oxygen, and eliminate enough carbon dioxide [1]. On the other hand, the tidal volume should not be excessive as it may damage the alveoli due to the large volumes and high pressures (volutrauma and barotrauma) [2].

In pressure modes, the clinician controls the tidal volume by setting the inspiratory pressure (either a peak inspiratory and an end-expiratory pressure or a delta pressure on top of the end-expiratory pressure). The ventilator displays the delivered inspiratory and the received expiratory tidal volume that it measures, at the pressure set by the clinician [3]. When using the very popular volume-guaranteed modes, an algorithm controls the pressure to reach a volume that is set by the clinician [4]. Whichever mode is being employed, an accurate measurement of tidal volume is essential to control the delivery of an optimal tidal volume.

### 2.1. How Ventilators Measure Tidal Volumes

There are four major factors to consider when interpreting the measured tidal volumes that are reported by the ventilator: (1) the phase during which the tidal volume is measured (inspiration versus expiration); (2) the site of tidal volume measurement within the breathing circuit; (3) the accuracy of the compliance compensation; and (4) the conditions under which tidal volume is being measured. We will discuss all four factors and how ventilators handle these factors in their tidal volume measurement and reporting. Even though they are very relevant for accurate and optimal ventilation, flow mechanics and dead space are not within scope of this article.

First, tidal volume is typically defined as the volume of gas that is being inspired and expired with each breath. However, the bidirectionality in its definition makes it already obvious that there is no single exact definition of a tidal volume. In pediatric ventilation, it is common to have a discrepancy between the inspired tidal volume and the expired tidal volume in a single respiratory cycle (mainly due to leakage of air from the use of uncuffed tubes). As the clinician sets the inspiratory pressure in pressure modes, the ventilator simply has to display both volume measurements. It is then up to the clinician to estimate how much volume is being lost during inspiration and during expiration, and adjust the inspiratory pressure. In volume-guaranteed modes the manufacturer has to make a choice which measurement of tidal volume to use as the reference value for the ventilator to adjust the inspiratory pressure.

The choice of measurement comes with assumptions. In the adult and pediatric setting, most ventilator modes use the inspiratory tidal volume as the reference volume. This choice makes the underlying assumption for controlling tidal volumes that there is no leakage and that (almost) all of the inspiratory tidal volume reaches the alveoli. Contrastingly, in the neonatal setting, ventilators use the expiratory measurement to control the pressures and tidal volume, which assumes that all leakage occurs during the inspiratory phase and that the volume that is measured at expiration is the full volume that went into the patient. Alternatively, some ventilators allow the clinicians to choose which tidal volume measurement should be used to set inspiratory pressures. The differences in the underlying assumptions between the different ventilators is reflected by the default tidal volumes in volume-guaranteed modes. In modes that use expiratory tidal volume the default is typically 4–5 mL/kg (i.e., in neonatal settings) [4] whereas it is typically 6–8 mg/kg that use the inspiratory tidal volume (i.e., adult and pediatric settings).

Second, the choice of which phase of the respiratory cycle to use as a reference go hand in hand with the site of measurement. Most ventilators measure tidal volumes within the machine both at the inspiratory and expiratory limb, and use one of them as the reference measurement. In contrast, some ventilators (typically used in neonatal settings) use a separate sensor between the Y-piece of the breathing tubes and tube connector at the patient side to measure tidal volumes [5]. Measurements of tidal volume within the machine or within the tubing system are systematically different [6]. These differences are caused by the breathing system’s compliance and the physical conditions of the measurements. In the following paragraphs, we will discuss these factors in more details, but the implication is that it is important to know where tidal volume is being measured to fully understand the displayed tidal volume.

Third, gas compression and the distensibility of the breathing circuit result in a loss of volume when the system pressurizes to deliver a breath. During its start-up procedure, a ventilator can measure the total compliance of the system. The measured compliance can then be used to estimate which tidal volume is necessary to compensate for the loss of volume and deliver the target tidal volume to the lungs of the patient. Inaccurate compliance compensation is much more common in paediatric patients than adult patients.

The clinical impact of compliance compensation in adults is almost none. However, in pediatrics it may be significant.^1^ A normal compliance of the breathing circuit is 0.2–1.5 mL cm H_2_O^−1^. When a patient is ventilated with a pressure difference of 20 cmH_2_O and a compliance of 1 mL cm H_2_O^−1^, the compensation is 20 mL. For an infant of 5 kg with a target tidal volume of 30 (6 mL kg^−1^), that is an error of 66% (or 4 mL kg^−1^). Even when applying weight-specific tubing and other components, the ratio between the compressible volume and the tidal volume is higher in smaller patients. This results in less accurate estimation of the volume needed to compensate the compliance of the breathing circuit than in larger patients [1].

Fourth, gas volume depends on the temperature and humidification in relation to the pressure. During inspiration, the tidal volume transitions from a dry gas at the ambient temperature and pressure of the breathing circuit (the ATPD (Ambient Temperature Pressure Dry) condition) to a water-saturated gas at the temperature and pressure of the patient’s lungs (BTPS (Body Temperature Pressure Saturated)) Consequently, the tidal volume expands during this transition, and thus the conditions in which tidal volume is measured do not necessary reflect the final volume that reaches the lungs.

### 2.2. How Ventilators Report Tidal Volume

With the combined knowledge of all of these four factors affecting the tidal volume measurement, it is possible to infer the alveolar tidal volume from the measured tidal volume using computational algorithms. In the past, most ventilators measured tidal volumes in the inspiration circuit and reported the tidal volumes “as is”, which means the ATPD condition. Knowing that the volume in the patient is expanded under BTPS conditions, the vendors of medical ventilators upgraded their software to display (more accurately) the tidal volume under BTPS conditions in the patient’s lungs. A schematic overview of ATPD versus BTPS is provided in Figure 1.

At sea level, 20 degrees Celsius and ambient humidity, the typical BTPS correction factor is 1.12. The formula to calculate this correction factor is shown in Formula (1). A correction factor of 1.12 means that the expected tidal volume under BTPS conditions is 12% larger than the tidal volume under ATPD conditions. At 6–8 mL kg^−1^ this 12% increase reflects a 0.7–1.0 mL kg^−1^ increase in tidal volume, which likely has a clinical impact.

Besides applying an algorithmic correction, another solution is to have heated flow sensors. Multiple ventilators [7] have heated flow sensors, but most of them only at the expiratory limb and not at the inspiratory limb. However, heated flow sensors [8] also have technical issues [9]. Air flow cools the measurement environment, but not in a consistent way due to flow dynamics. For example, flow dynamics play a role in forced expiratory volume in 1 s (FEV_1_) measurements [8]. However, forced expiration is different from passive expiration during pediatric mechanical ventilation, so the flow dynamics are, within reason, unlikely to be of clinical impact. Consequently, most ventilator manufacturers have thus focused in the last decade on advancing their software by increasing their technical knowledge to better inform clinicians of an accurate tidal volume and to limit ventilator-induced harm to patients.
(1)$$\frac{V\mathrm{btps}}{V\mathrm{atpd}}=\frac{T\mathrm{body}}{T\mathrm{amb}}\times \frac{P\mathrm{amb}-Ph2o\left(T\mathrm{amb},H\mathrm{amb}\right)}{P\mathrm{amb}-Ph2o\left(T\mathrm{body},H\mathrm{body}\right)}$$

The corrected tidal volume under BTPS conditions from ATPD volume measurements, dependent on temperature (T), partial pressure (P) and humification (H) [10].

## 3. Pediatric Ventilation Tidal Volume: Impact of a Software Upgrade in a Prospective Cohort Study

The impact of a software upgrade with respect to ATPD and BTPS conditions on the tidal volumes can be illustrated by a prospective evaluation at the PICU of the University Medical Centre Utrecht, The Netherlands between 2012 and 2014. Getinge (at that time Maquet) upgraded the SERVO-i^®^ ventilators (Getinge, Solna, Sweden) to software version 7.0 and the only option was to start using BTPS (before software version 7.0 Getinge was using ATPD) [11]. With the calculated impact in mind, the switch between ATPD and BTPS was performed on one day for both SERVO-i^®^ ventilators and the Engstrom ventilators of GE Healthcare, as they had the option to choose between ATPD and BTPS.

### 3.1. Data Collection

For this study, all children admitted to the PICU of the University Medical Centre Utrecht during 2012–2014 and ventilated with a Maquet SERVO-i^®^ ventilator for >24 h on Pressure Regulated Volume Controlled mode were included. As there were technical issues with humidity and the flow sensor of the Engstrom ventilators during the BTPS period (this issue was independent of the software upgrade) infants were predominantly ventilated with the SERVO-i^®^.

Deidentified patient data were acquired retrospectively. Demographic data consisted of age in months and weight in kilograms. Per minute ventilator data consisted of set inspiratory tidal volume (ml kg^−1^), respiratory rate (rate min^−1^), peak pressure (cm H_2_O) and minute volume per weight (ml kg^−1^). Per minute monitor data consisted of body temperature (degrees Celsius) and end-tidal CO_2_ (mm Hg). These data were collected one year before the software update (2012–2013, ATPD ventilated patient cohort) and one year after the software update (2013–2014, BTPS ventilated patient cohort). Statistical analysis to compare both cohorts included Student’s T-test for continuous values and was performed using R software version 3.4.2 (R Foundation for Statistical Computing, Vienna, Austria). The medical ethics board of the University Medical Centre Utrecht, The Netherlands waived the need for informed consent.

### 3.2. Results

A total of 454 patients produced over 1 million observations. The median age was lower in the BTPS period than in the ATPD period (11 months versus 4.3 months), which can be explained by selectively using the SERVO-i^®^ in infants during the BTPS period. The median tidal volume in all patients was 7.29 mL/kg (Table 1), where the difference in median tidal volume was clinically insignificant (BTPS 7.27 mL kg^−1^ versus ATPD 7.31 mL kg^−1^). A clinically relevant difference was observed in the peak pressure (BTPS 20.2 cm H_2_O versus ATPD 21.8 cm H_2_O). This 8% difference in peak pressure in PRVC mode can be explained by the fact that the air in expands when transitioning to the BTPD condition in the lungs. Thus, when the volume expansion is included in the reference measurement for tidal volume, the target tidal volume will be reached at lower pressure levels. The observed difference of 8% matches the 8% difference that was described in the introduction document provided by the manufacturer Getinge at the time of BTPD introduction.

### 3.3. Discussion of the Results

The major limitation of this illustrative prospective evaluation is that it was impossible to adjust for all confounding due to the observational nature. First, there was a difference in patient population between the two periods, as coincidental circumstances resulted in using the SERVO-i^®^ predominantly in infants. Nonetheless, the mean tidal volumes per kilogram were similar between groups. One might expect that the same tidal volumes would be reached at the same pressures, regardless of age. Thus a 1.6 cm H_2_O difference is substantial. Second, the pulmonary compliances (reflecting the degree of pulmonary illness) were not available for further analysis. So even when we would be able to adjust for differences in age, we would still not know for sure whether differences in underlying illnesses had a confounding effect as the case mix of infants is different from older children (e.g., toddlers). Hence, we simply presented crude results as an example, and the prospective evaluation should only be seen as supportive of the narrative review. Still, this software change is currently active and represents the most important software change with respect to the interpretation of clinical measurements in the past decade.

In conclusion, this software change did have an impact on data in clinical care in this example study. It was not as much as the theoretically estimated 12% change in volume, but instead it resulted in an 8% decrease in peak pressures (as was predicted by the vendor). However, the clinical impact on patient outcome (e.g., on baro- or volutrauma) is hard to establish due to the heterogeneity of the population and the absence of objective ventilator outcome measures.

## 4. Summary

The holy grail of a tidal volume of 6 mL/kg in protective adult ventilation [12] includes a major assumption: the measurement of tidal volume is uniform and indisputable. However, technical advancements in ventilators and their software, combined with the prominent challenges of, e.g., leakages in paediatric ventilation which demand a better understanding of the impact of these issues in the paediatric intensive care population (especially in infants and toddlers). Multiple technical choices made by the vendors of the ventilators may have an impact on actual lung ventilation with respect to tidal volumes and the peak pressures required to achieve them [13]. Such an impact may be relatively insignificant in adults, but may have potential clinical relevance in the paediatric population.

Improved understanding and knowledge of the technical choices in ventilator setups, defaults, and algorithms has major potential to improve the daily clinical PICU practises in ventilation. Equally important may be the need to include this technical setup in research papers. All described components in this expert overview have the potential to alter the tidal volume measurement with at least 10%. In our search for a tidal volume range that can be considered the holy grail for lung-protective ventilation, we should be sure we are comparing the same tidal volume, measured at the same phase, at the same site and under similar conditions.

## Figures and Tables

**Figure 1 jcm-11-00098-f001:**
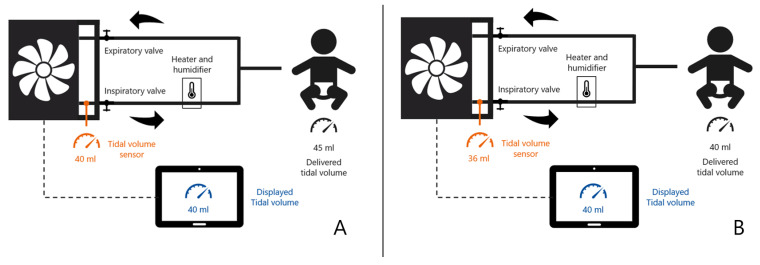
Schematic example of a ventilator displaying tidal volume (V_T_) under ATPD (**A**) versus BTPS (**B**) conditions in a ventilated infant. In (**A**), inspiratory V was sensed and displayed as 40 mL, however, after heating and humidifying of the gas, the actual delivered V_T_ was 45 mL. In (**B**), inspiratory V_T_ was sensed as 36 mL, but after a (software) correction displayed and delivered as 40 mL.

**Table 1 jcm-11-00098-t001:** Differences between ATPD ventilated patients and BTPS ventilated patients.

	All Patients	ATPD Ventilated Patients	BTPS Ventilated Patients	*p*-Value
	*n* = 454 ** (1,063,901 observations)	*n* = 221 (532,930 observations)	*n* = 235 (551,937 observations)	
Age, median (IQR), months	5.6 (28.1)	11.3 (63.0)	4.3 (9.6)	<0.001
Weight, median (IQR), kg	6.1 (8.3)	8.4 (15.3)	5.45 (4.7)	<0.001
Body temperature, mean (SD), degree Celsius ***	37.0 (0.8)	36.9 (0.8)	37.0 (0.8)	<0.001
Tidal volume per kg, mean (SD), mL/kg *	7.29 (1.36)	7.31 (1.32)	7.27 (1.39)	<0.001
Respiratory rate, mean (SD), /min *	36 (7)	35 (8)	37 (6)	<0.001
Peak pressure, mean (SD), cm H_2_O *	21.0 (5.6)	21.8 (5.4)	20.2 (5.6)	<0.001
Minute Volume per kg, mean (SD), mL/kg	264 (75)	259 (83)	267 (66)	<0.001

* analysis of all observations per ventilator. ** some patients were admitted in both ATPD and BTPS periods. *** available in 827,594/1,063,901 observations. Abbreviations: ATPD—Ambient Temperature Pressure Dry; BTPS—Body Temperature Pressure Saturated; IQR—Interquartile Range; SD—Standard Deviation.

## Data Availability

The datasets analyzed during the current study are available from the corresponding author on reasonable request.

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
