# Peer review of "Tidal Volume in Pediatric Ventilation: Do You Get What You See?"

_jcm, 2021, doi:10.3390/jcm11010098_

Round 1

Reviewer 1 Report

This article, although the results are not very striking, emphasise the importance of the measurement technique of the Tidal volume.  I have some comments about this article: 

- The content and data are preliminary and  observational.

  • As the authors mentioned in the limitation section, the study population is heterogeneus to conclude. Second, effect of underlying disease was neglected.  In this context, sample size is questionable.
  •  In the Table 1 median age (IqR) values   were given  twice, and in the text mean ages were given. This is confusing. 
  • Adjustments for not all the factors, but  age should be performed.
  •  Did the authors measure inter-individual and intra-individual differences in TV and PIP levels? The results taken from the comparison between the two groups may  meaningful only if the differences were greater than intra-individual differences.
  • Almost ten years pas after this study and I would like to understand whether there has been any change in the ventilatory technology.

Manuscript was generally written like a review article rather than original research. Due to the design and methodological limitations,  it does not meet the criteria of original research.   In my opinion, this article is considerable in the category of letter to the editor or short reports. 

Author Response

Dear reviewer see rebuttal in Word

Reviewer 2 Report

I advise the authors to better explain the paragraph on methods, but above all to enrich the discussion with data from the literature by deepening the possible damage associated with incorrect ventilation. For example, in newborns and infants, the use of high tidal volumes contributes to the development of oxidative stress with a consequent risk of brain injury.

Author Response

Dear reviewer see Word document

Round 2

Reviewer 1 Report

I've read the replies to reviewers' comments. The article category is the responsibility of the editors, as they are invited authors. Other comments and corrections in the text are appropriate.